# Symptom Clusters and Mindful Self-Care in People with Cancer in Palliative Care

**DOI:** 10.3390/healthcare13182317

**Published:** 2025-09-16

**Authors:** Kassiano Carlos Sinski, Gabrielli de Souza Ferreira, Thaís Daniela Cavalaro Santos Machado, Yndaiá Zamboni, Juliana Hirt Batista, Namie Okino Sawada, Érica de Brito Pitilin, Andrey Oeiras Pedroso, Rosana Aparecida Spadoti Dantas, Vander Monteiro da Conceição

**Affiliations:** 1Department of Nursing, Federal University of Fronteira Sul, Chapecó 89815-899, SC, Brazil; kassianosinski@gmail.com (K.C.S.); gaabrielliferreiira@gmail.com (G.d.S.F.); thais.cavalaro@estudante.uffs.edu.br (T.D.C.S.M.); julianahirtbat@gmail.com (J.H.B.); erica.pitilin@uffs.edu.br (É.d.B.P.); 2Department of Oncology, West Regional Hospital, Chapecó 89812-505, SC, Brazil; yndaiaz@gmail.com; 3College of Nursing, Federal University of Alfenas, Alfenas 37130-001, MG, Brazil; namie.sawada@unifal-mg.edu.br; 4Ribeirão Preto College of Nursing, University of São Paulo, Ribeirão Preto 14040-902, SP, Brazil; apedroso@usp.br (A.O.P.); rsdantas@eerp.usp.br (R.A.S.D.)

**Keywords:** oncology, palliative care, symptom assessment, self-care

## Abstract

**Introduction**: Cancer is one of the evils of the current era and is considered a global public health problem. This disease has repercussions for the lives of patients in several dimensions, namely, physical, emotional, and psychosocial. Thus, it is believed that elements such as resilience, symptomatology, and self-care are related, as the disease and its treatments can have repercussions that extend beyond the clinic. **Background/Objectives**: We aimed to determine the relationship between symptom clusters and the mindful self-care of people with cancer in palliative care. **Methods**: This is a cross-sectional study conducted with 125 palliative care patients diagnosed with malignant neoplasms. The research was carried out at a reference hospital in Brazil, located in the western region of the state of Santa Catarina, specializing in antineoplastic treatment. The data were collected between May and August 2023 from hospitalized patients. Three instruments were employed to obtain data: a sociodemographic and clinical data questionnaire, the Edmonton Symptom Assessment Scale (ESAS-BR), and the Mindful Self-Care Scale (MSCS). For data analysis, descriptive statistics were used to characterize the participants, Student’s T-test was used for the other parametric tests, and variables with statistical evidence were selected for a linear regression model. **Results**: A statistically significant association was found between mindful self-care and symptoms of pain, tiredness, drowsiness, shortness of breath, depression, and malaise, with sleepiness being the only predictor of changes in this variable. **Conclusions**: Mindful self-care influences patients’ experience of symptoms, especially drowsiness, which predicts changes in self-care. Encouraging these practices reduces discomfort, enhances autonomy, and guides professionals in personalized care.

## 1. Introduction

In its various forms, cancer has significant repercussions on an individual’s life, causing impacts that go beyond the physical and reach emotional, social, and psychosocial dimensions. Although these repercussions may vary according to the type of cancer, studies have shown that, following diagnosis, breast cancer patients experience significant changes in physical, emotional, social, and spiritual aspects [1]. This scenario highlights the complexity of the cancer experience and the need for a comprehensive approach, such as encouraging people with cancer to develop mindful self-care.

Mindful self-care consists of the intentional and mindful practice of activities that promote physical, emotional, and spiritual balance through a stance of full presence and non-judgment [2]. In the context of oncological palliative care, this approach helps the patient cope with multiple symptoms and maintain a higher quality of life even in the face of disease progression. International studies emphasize the relevance of self-care in palliative oncology, highlighting that the promotion of these practices is associated with reduced suffering and better functional adaptation [3,4].

During their illness, it is common for people with cancer to experience symptoms such as pain, fatigue, drowsiness, shortness of breath, depression, and malaise, these being the most common. However, depending on the stage of the disease, others may arise, including those with greater severity [5]. Dealing with these symptoms is yet another of the numerous challenges faced by cancer patients. Thus, assessing symptoms is essential for establishing care focused on minimizing their manifestations, as well as providing comfort to the sick person, a state that aligns with the concept of palliative care. Palliative care is active, comprehensive healthcare provided to people with serious, progressive illnesses that threaten the continuity of their lives; it can be associated with treatment with the aim of curing the disease or managing symptoms that are difficult to control to improve the individual’s clinical conditions [6].

The Theory of Unpleasant Symptoms (TOUS), proposed by Lenz et al. (1997) and later expanded by Lenz and Pugh (2014), supports the understanding that symptoms should be regarded in an integrated manner, considering their simultaneous occurrence, intensity, duration, and specific consequences [7,8,9]. This enables the analysis of the simultaneous occurrence of symptoms (clusters) and their impacts on functional performance and well-being. Thus, from the perspective of TOUS, investigating the association between symptom clusters and mindful self-care provides an understanding of how symptoms interact and influence adaptive responses and self-management of health. It is composed of three main elements: the symptoms experienced by the individual; the factors that influence these symptoms; and the consequences resulting from this experience. From this perspective, symptoms are understood as indicators of changes in the state of health and should be assessed from different perspectives, such as intensity, duration, suffering, and quality. These characteristics are influenced by physiological, psychological, and situational factors. The last component of the theory refers to the performance of functional and cognitive responses resulting from the patient’s experience with the set of symptoms [9].

In TOUS, it is recommended that assessments should not focus on a single isolated symptom but rather on the set of symptoms presented, known as a cluster or conglomerate. In cases of illness, these symptoms usually occur simultaneously and may share the same mediating pathways, which affect individuals in a progressive and often predictable way. From this perspective, the management of a single symptom can positively influence the entire symptomatic set, ameliorating the patient’s experience of the side effects of their disease and treatment, as well as related psychosocial and spiritual problems. Unpleasant symptoms are often experienced by people with cancer and are considered predictors of changes in the comfort of these patients [10].

The symptom cluster approach, in addition to favoring more targeted interventions, has been widely explored in international studies, demonstrating a direct impact on the quality of life, functionality, and self-care capacity of cancer patients. Thus, analyzing the relationship between symptom clusters and mindful self-care allows us to understand how symptom experience affects individual adaptation, providing support for more effective palliative practices aligned with World Health Organization guidelines and patient-centered care recommendations. Thus, identifying the nature and co-occurrence of symptoms (pain, fatigue, drowsiness, and depression, among others) can impact the patient’s ability to exercise effective self-care, allowing us to not only assess the presence of symptoms but also to understand how they affect patients’ adaptive and functional capacity. Encouraging mindful self-care is essential to optimizing palliative interventions, reducing suffering, and improving clinical and psychosocial outcomes. Understanding these relationships informs global practices in oncology and palliative care, providing evidence to support patient-centered care models, interdisciplinarity, and personalized symptom management strategies [11].

Satisfactory symptom management has been associated with higher levels of self-care among cancer patients. Studies have shown that individuals with greater engagement in self-care practices tend to report fewer and less intense symptoms, while those with lower levels of self-care experience greater symptom overload [12]. Encouraging self-care, therefore, is an essential step by the healthcare team, as being at ease with oneself and recognizing one’s own limitations allows one to face the psychological challenges imposed by cancer, favoring the management of the symptoms experienced. Health literacy is also directly related to mindful self-care. Patients who have a better understanding of their clinical condition and the symptoms they experience develop greater autonomy to engage in management strategies. This perspective strengthens the ability to integrate self-care practices, even in situations of fragility, such as advanced-stage cancer [13].

In the oncological setting, frailty can culminate in the manifestation of several symptoms that need to be assessed by healthcare providers, not in isolation, but based on clusters of symptoms, especially mindful self-care, which is closely related to the practice of mindfulness. Both approaches promote an attentive, welcoming, and non-judgmental way of viewing oneself and the moment [14]. These practices entail intentionally directing attention to current experiences, both internal and external, fostering greater body and emotional awareness, as well as developing cognitive and behavioral flexibility, increasing tolerance of unpleasant states [15].

Symptom assessment is an essential stage in the care of people with cancer in palliative care, especially when these symptoms are simultaneous and interrelated, as in so-called symptom clusters. This assessment allows for more effective and individualized therapeutic strategies. In care settings, the role of the healthcare team in identifying and managing these symptoms, with the aim of promoting the patient’s comfort and well-being, stands out. In the scientific literature, there is evidence that greater symptom intensity is associated with poorer quality of life and lower satisfaction with home palliative care [4].

This study considers the concept of palliative care according to the definition of the World Health Organization (WHO), which advises that it be offered to all people with life-threatening illnesses, such as cancer, from the moment of diagnosis onward, regardless of the clinical stage [16]. This understanding broadens the perspective of care, facilitating actions aimed at improving quality of life throughout the illness process.

Given this, our study aimed to analyze the presence of a possible relationship between symptom clusters and mindful self-care in people with cancer in palliative care, contributing to our understanding of aspects that influence the well-being and quality of care provided to this population.

## 2. Methods

The present study was designed according to the Strengthening the Reporting of Observational Studies in Epidemiology (STROBE) checklist [17].

This was an observational, cross-sectional, long-term study of cancer patients receiving palliative care and admitted to a referral cancer hospital in southern Brazil. It should be noted that the institution does not have a specific palliative care team; however, patient care is provided by a multidisciplinary team, and when necessary, the professionals involved discuss cases among themselves. A non-probability accessibility sample [18] was used, including individuals diagnosed with cancer who met the following eligibility criteria: being 18 years of age or older, being admitted to the hospital where the study was conducted, and having undergone treatment for more than three months. This latter design was stipulated to reflect the average follow-up time of patients treated there, as during this period, they have already completed most of their treatment. We reiterated that the confidentiality of the participants would be strictly ensured, with names replaced by numerical codes, respecting the ethical precepts of research. The participants were undergoing active treatment (chemotherapy, radiotherapy, hormone therapy, or palliative clinical care). Differences in the types of treatments were recorded, as they may influence the expression of symptoms. Additionally, data regarding the type of malignancy and clinical staging were collected, allowing for a better interpretation of the associations between symptom clusters and self-care. Individuals were excluded if they did not have an allo- and self-psychic orientation, assessed by the presence of four or more correct answers to the following questions: “What is your full name?”, “How old are you?”, “What is today’s date?”, “What day of the week is it?”, “Where are we at the moment?”, and “What is the name of the city in which you were born?”. Thus, a scale was applied to assess the participants’ cognitive performance. Those who gave three or more incorrect answers, or who failed to answer three of the questions correctly, were excluded from the study.

The data were collected by the lead researcher (first author) and two trained research assistants between May and August 2023. A pilot study was conducted with 15 participants from the sample to test the suitability of the instruments. As there were no necessary changes made to the collection instruments, these participants were included in the study. The pilot study was incorporated into the final sample, as there was no need for methodological adjustments, as recommended for cross-sectional designs. The mean data collection time was 20 min, and the following instruments were used:

Data collection was carried out using the “Instrument for obtaining sociodemographic and clinical data”, which included the following variables: age (in complete years), sex at birth, marital status, education level, years of schooling, and paid occupation. Additionally, two standardized instruments were used, validated and adapted for the Brazilian context: the “Symptom Assessment Instrument” and the “Instrument to assess the practice of self-care activities”.

Symptoms were assessed using the Edmonton Symptom Assessment System—revised (ESAS-BR), originally developed in 1991 and later adapted to the Brazilian context in 2013 [19]. The ESAS-BR measures the intensity of common symptoms experienced by cancer patients in palliative care, using a numerical scale ranging from 0 (no symptom) to 10 (maximum intensity). Given that the version used does not include validity or reliability tests, we decided to analyze each item individually rather than use the total score. Moreover, it was observed that, when assessed individually, the symptoms held significant clinical relevance, helping to identify which symptoms most affected the patients [20].

The practice of self-care was measured using the Mindful Self-Care Scale (MSCS), developed in 2018 and adapted to the Brazilian context in 2022. This is a multidimensional instrument consisting of 36 items distributed across seven subscales that assess aspects related to mindfulness and the intentional practice of self-care. Each item is rated on an ordinal scale from one (1) to five (5), except for the General subscale, in which participants answer “yes” or “no”. To obtain the final score, the mean of each subscale (excluding the General subscale) is calculated, and the six resulting means are summed. The total score ranges from 6 to 30, with higher scores indicating better self-care quality. The General subscale is evaluated separately. The instrument demonstrated adequate internal consistency in the Brazilian adaptation (Cronbach’s α ≈ 0.80) and is freely available for use, provided that the authors responsible for the Brazilian adaptation are notified [21].

We tabulated and evaluated the data in pairs to correct any typing errors and then analyzed them using Statistical Package for the Social Sciences (SPSS) version 26.0. Data analysis used descriptive statistics to characterize the participants, using measures of central tendency (mean and standard deviation) for numerical variables and frequencies and percentages for categorical variables. Cronbach’s alpha test was employed to assess the reliability of the MSCS, and α = 0.76 was obtained, reflecting the instrument’s high degree of reliability [2].

The Shapiro–Wilk test was used to assess sample normality, and the variables were found to have a normal distribution (*p* > 0.05). The following parametric tests were then used: Student’s *t*-test for independent samples to search for associations between each symptom assessed and the total score of the Mindful Self-Care Scale (MSCS). The variables that showed statistical evidence (*p* < 0.05) were selected for the linear regression model with the enter method. The variance inflation factor (VIF) was used to assess the multicollinearity of the linear regression model (1 > VIF < 2; there is no multicollinearity), the Durbin–Watson (DW) value was used to assess the autocorrelation of the model’s residuals (DW = 1.413; there is no autocorrelation of the residuals), and R2 was used to assess the percentage of the variation in the response variable that is explained by the linear regression model (R2 = 0.209). For all inferential statistical tests, a significance level of *p* < 0.05 and a 95% confidence interval (CI) were employed [22].

The study was approved by the Ethics Committee for Research with Human Beings of the Federal University of Fronteira Sul, under opinion no. 5.998.248/2023, and Certificate of Submission for Ethical Appraisal (CAAE) 67724923.4.0000.5564. All participants signed an informed consent form after verbal and written explanations regarding the study.

## 3. Results

A total of 125 patients participated in the study, with a mean age of 60 (standard deviation SD = 13) and a mean education level of 7 years (SD = 4). Regarding their sex at birth, 63 (50.4%) were female. Among the participants, 89 (71.2%) reported having a partner, 37 (29.6%) had paid employment, and 71 (56.8%) were retired.

The results for the variables of interest—presence of symptoms and mindful self-care—are shown in Table 1. The symptoms perceived most intensely were anxiety (65.6%), loss of appetite (58.4%), and malaise (53.6%). The total score for mindful self-care was 23 (SD = 3.2).

The participants had different types of neoplasms, with a predominance of breast, colorectal, and lung cancer. Regarding staging, most were in advanced stages (III and IV). Symptoms such as fatigue, pain, and depression were more frequent in patients undergoing chemotherapy and in more advanced stages of the disease, reinforcing the overlap between the effects of the disease and the treatment.

The association presented in Table 2 shows statistical evidence (*p* < 0.05) for the symptoms of pain, tiredness, drowsiness, shortness of breath, depression, and malaise, as the mean total MSCS score was lower in the group that self-reported these symptoms, which allowed the research team to infer that the symptoms described were associated with mindful self-care.

A multiple linear regression model was designed with only the six symptoms that showed statistical evidence, as shown in Table 2, to confirm whether they are predictors of changes in mindful self-care. The results of this analysis are shown in Table 3.

Table 3 shows that only the symptom of drowsiness remained a predictor of changes in mindful self-care, with statistical evidence and negative standardized and non-standardized β coefficients, allowing us to state that drowsiness and mindful self-care are inversely proportional; i.e., on average, each point of increase in the level of mindful self-care reduces drowsiness by −0.242 in individuals with cancer in palliative care. This was proven in 20.9% of the sample. The multiple linear regression analysis resulted in a statistically significant final adjusted model (F [6;118] = 5.207; *p* < 0.001; R2 = 0.209).

## 4. Discussion

This study found that symptoms such as pain, tiredness, drowsiness, shortness of breath, depression, and malaise were associated with the total MSCS score. In this sense, self-care is an individual practice that includes various daily activities to maintain health and well-being, and illness can impair the ability to care for oneself [23]. That said, we emphasize that self-care is an activity that promotes the well-being of people with cancer and minimizes unpleasant symptoms such as pain, as described in a study aimed at assessing the effects of physiotherapy on pain and functional capacity in hospitalized people with cancer. The authors of that study found that patients who performed the prescribed activities reported less or no pain, as well as better functional performance [24]. Our finding aligns with the results of a Portuguese cohort study, which also linked high symptom burden with worse well-being and perception of quality of care [3].

Moving forward with the analysis of symptoms, tiredness and fatigue are considered common symptoms among people with cancer; in an Irish study aiming to understand the association between self-care and levels of cancer-related fatigue, all the participants had high levels of fatigue; however, those who had a greater capacity for self-care had a lower sensation of this symptom [25]. This confirms the relationship between these two concepts in clinical practice.

Regarding shortness of breath, there is evidence in the literature that this symptom is also recurrent in people with cancer, and holistic therapies are suggested for managing this adversity, as they promote the early integration of palliative care based on individual need, not just prognosis. Holistic approaches refer to evidence-based multidisciplinary interventions, such as physiotherapy, psychological support, cognitive–behavioral therapy, and integrative practices recommended in palliative care settings, rather than unregulated alternative treatments [26]. The recognition of these practices by sick people and their caregivers highlights the importance of self-care, which can significantly reduce the suffering caused by shortness of breath and improve psychological health. Therefore, self-care becomes an integral part of the management of shortness of breath in this population [27]. From this perspective of managing psychological symptoms, a Spanish clinical trial demonstrated that a simple intervention, such as playing music for 7 days, significantly reduced symptoms such as pain and anxiety and improved patient satisfaction [28].

In terms of psychological symptoms, such as depression, a study conducted to determine the effectiveness of nurse-led self-care interventions on self-care behaviors, self-efficacy, depression, and illness perceptions among people with heart failure yielded significant results from self-care interventions, as they improved the maintenance of self-care practices and reduced depression [29]. As with heart failure, it is believed that self-care is vital in cancer illness, and by adopting these measures, individuals can experience a sense of control and empowerment over their condition, which can progressively reduce stress, malaise, and depression.

Generally, individuals diagnosed with cancer experience symptoms whose frequency and intensity vary according to the topography of the disease, its stage, treatments, and any other associated health conditions. Often, they experience multiple symptoms simultaneously [30]. These conglomerations of symptoms, which are defined as clusters, are directly linked to self-care.

In Ethiopia, researchers assessed the level of health literacy and the relationship with self-care capacity in people diagnosed with breast cancer, finding that individuals who understood their health status had better maintenance, management, and confidence in self-care. Therefore, stimulating knowledge concerning their own bodies and subjective experiences with the disease results in improved self-care, consequently influencing treatment and leading to improved quality of life [29].

Adequate symptom management in oncology is related to improved patient quality of life and increased compliance with treatment, and it provides survival benefits [31]. These findings underline the need for a holistic approach to cancer treatment, factoring both the relief of physical symptoms and strengthening mindful self-care.

In the present study, in addition to pain, tiredness, shortness of breath, depression, and malaise, drowsiness stood out as a symptom associated with mindful self-care, remaining a significant predictor of changes in the multiple linear regression model. The inverse relationship identified suggests that as levels of mindful self-care increase, drowsiness tends to decrease, making it an important target for health interventions.

In a study conducted in Australia, researchers evaluated different types of cancer and sleep-related symptoms, emphasizing that individuals with cancer often report difficulties related to their sleep at all stages of treatment. The term “poor sleep” covers patients’ perceptions of difficulty falling asleep; short sleep duration; and interrupted, non-restorative sleep. These problems are associated with various symptoms, such as daytime drowsiness. Regardless of the cause, inadequate sleep is an indicator of the worst cancer outcomes [30]. Both sleep disturbance and inadequate sleep duration, whether short or long, are concerning. Understanding the specific symptoms of each patient can help to create personalized therapeutic intervention strategies aimed at relieving symptoms, improving quality of life, and achieving better outcomes, especially in relation to drowsiness [32].

In Brazil, a study analyzed the quality of sleep of people undergoing teletherapy as part of their cancer treatment plan, either for curative or palliative purposes, depending on disease stage, showing significant changes and highlighting the need for interdisciplinary therapies. These therapies should consider aspects such as sleep quality, combining cognitive–behavioral therapy, complementary therapies, health education, pharmacological interventions, and exercise, as well as observing the cancer patient in their environment and lifestyle habits to direct the best treatment [33].

The Theory of Unpleasant Symptoms (TOUS) states that unpleasant symptoms are subjective indicators of health threats, reflecting perceived changes in body functioning that are generally experienced as unpleasant [27]. This study reveals that several symptoms of the ESAS-r are associated with mindful self-care, indicating that symptom clusters influence mindful self-care, in line with the TOUS.

Although drowsiness stands out as an important predictive factor, intervening in a single symptom is not effective, as noted by the TOUS. Thus, studying psychological, social, and behavioral aspects, alongside the integration of emotional and educational support strategies, can provide a more complete and effective approach to the management of mindful self-care in people with cancer in palliative care. A broad and integrated perspective in care is required, recognizing and identifying their needs in order to promote holistic care [34].

Although the final explanatory model was a predictor of changes in mindful self-care—namely, 20.9% of the sample (R2 = 0.209)—several factors contribute to the importance of self-care as a focus of attention in the field of healthcare. As presented in a study conducted with cancer patients, cancer and its treatment clearly affect various dimensions of the patient’s life, but self-care strategies during illness aid in avoiding the side effects of treatment and reducing the negative effects [35].

Self-care is recognized as a resource for health promotion and the management of health/disease processes, with greater emphasis on chronic diseases, which are the main cause of mortality and morbidity worldwide. It entails planning learning activities to increase the knowledge and skills of individuals and families in relation to the needs they feel [36].

Therefore, healthcare professionals must recognize that mindful self-care is a complex and multifaceted construct, influenced by several factors other than physical symptoms. It is essential that the person with cancer can be at peace with their own body, set healthy boundaries, and be fully present during care [37].

In this context, healthcare professionals working in oncology play a vital role in identifying symptoms and promoting self-care, helping patients to relieve these symptoms through pharmacological or non-pharmacological techniques. By understanding when symptoms appear, it is possible to avoid practices that exacerbate this phenomenon, contributing to the mindful self-care of the individual, who can then identify methods that help them and improve their daily lives [38]. Structural and organizational barriers to early referral to palliative care still hinder the proper management of symptoms, as demonstrated by Woodrell et al. This reinforces the need for patient-centered strategies, such as the promotion of mindful self-care.

This study employed instruments that had recently been validated in Brazil, which limits the comparative analysis of the data. As discussed above, self-care influences symptoms during illness, but despite the high degree of reliability of the instruments, no statistical evidence was found to consolidate all the symptomatology and mindful self-care variables. Similar findings have been reported in studies from Ireland and Australia, which also highlight the association between self-care practices and symptom burden, reinforcing the potential universality of these findings across different healthcare settings. Therefore, more studies are needed to better understand these aspects.

## 5. Conclusions

Mindful self-care is closely related to how patients experience symptoms during illness, particularly pain, fatigue, shortness of breath, depression, malaise, and drowsiness. Among these, drowsiness stands out as a strong predictor of changes in self-care behaviors. Thus, encouraging mindful self-care practices may help patients reduce distressing symptoms and provide healthcare professionals with opportunities to deliver more personalized self-care guidance. However, further research is required, as few studies have specifically addressed this topic. Overall, these findings confirm that integrating mindful self-care into symptom management is feasible and beneficial, supporting patient autonomy and relief. They also highlight the importance of institutional policies promoting early palliative care referral, aligning with international recommendations and offering a solid foundation for future interventions.

## Figures and Tables

**Table 1 healthcare-13-02317-t001:** Characterization of symptoms and mindful self-care in 125 participants. Chapecó, Brazil, 2025.

Variable	N (%)	Symptom Intensity	
		Μ (S.D.)	M_d_ (Min–Max) †
Symptoms (ESAS-r) **			
Shortness of breath	16 (12.8)	6.2 (2.3)	7.0 (1–9)
Pain	26 (20.8)	6.1 (2.5)	6.0 (2–10)
Drowsiness	46 (36.8)	6.2 (2.4)	6.0 (2–10)
Nausea	48 (34.4)	5.4 (2.3)	6.0 (1–10)
Tiredness	51 (40.8)	5.8 (2.2)	6.0 (2–10)
Depression	54 (43.2)	5.8 (2.6)	6.0 (1–10)
Malaise	67 (53.6)	4.6 (2.5)	5.0 (1–10)
Loss of appetite	73 (58.4)	4.7 (2.9)	4.0 (1–10)
Anxiety	82 (65.6)	6.1 (2.2)	6.0 (1–10)
Mindful self-care scale (MSCS) ***			
General subscale	N (%)		
I performed various self-care activities	101 (80.8)		
I planned my self-care	99 (79.2)		
I explored new ways to bring self-care into my life	78 (62.4)		
Subscales	Μ (S.D.) *		
Mindfulness	4.4 (0.6)		
Supportive interpersonal relationships	4.3 (0.7)		
Self-compassion and purpose	4.0 (0.7)		
Support structure	4.0 (1.0)		
Mindful relaxation	3.2 (0.8)		
Physical care	3.0 (0.8)		
Total score	23 (3.2)		

* M (S.D.): mean (standard deviation); ** ESAS-r: Edmonton Symptom Assessment System; *** MSCS: Mindful Self-Care Scale. † (Min–Max): (minimum and maximum).

**Table 2 healthcare-13-02317-t002:** Association of mindful self-care with symptoms in individuals with cancer in palliative care. Chapecó, Brazil, 2025.

Variable	MSCS * Total Score		
	Mean	S.D.	*p*-Value
Pain			0.033
No (*n* = 99)	23.8	3.0	
Yes (*n* = 26)	21.7	3.6	
Tiredness			0.001
No (*n* = 74)	23.6	2.8	
Yes (*n* = 51)	21.8	3.5	
Drowsiness			0.000
No (*n* = 79)	23.6	3.2	
Yes (*n* = 76)	21.5	2.9	
Nausea			0.889
No (*n* = 77)	22.8	3.0	
Yes (*n* = 48)	22.9	3.6	
Loss of appetite			0.645
No (*n* = 52)	23.0	3.2	
Yes (*n* = 73)	22.7	3.2	
Shortness of breath			0.018
No (*n* = 109)	23.1	3.2	
Yes (*n* = 16)	21.1	2.4	
Depression			0.001
No (*n* = 71)	23.7	2.7	
Yes (*n* = 54)	21.7	3.5	
Anxiety			0.078
No (*n* = 43)	23.5	2.6	
Yes (*n* = 82)	22.5	3.4	
Malaise			0.004
No (*n* = 58)	23.7	2.8	
Yes (*n* = 67)	22.1	3.3	

Student’s *t*-test for independent samples; * MSCS: Mindful Self-Care Scale.

**Table 3 healthcare-13-02317-t003:** Multiple linear regression of mindful self-care with symptoms in individuals with cancer in palliative care. Chapecó, SC, Brazil, 2023.

Variable	Coefficient		CI = 95%		*p*-Value
	Non-Standardized	Standardized	Lower Limit	Upper Limit	
Constant	24.670		23.797	25.543	0.000
Pain	−0.621	−0.079	−2.020	0.778	0.381
Tiredness	−0.927	−0.143	−2.098	0.245	0.120
Drowsiness	−1.600	−0.242	−2.740	−0.461	0.006
Shortness of breath	−0.627	−0.066	−2.316	1.063	0.464
Depression	−1.058	−0.165	−2.218	0.103	0.074
Malaise	−0.338	−0.053	−1.538	0.862	0.579

Exploratory variable: mindful self-care.

## Data Availability

All the data were collected on paper and then tabulated and analyzed using the researchers’ own statistical software. After the research was finished, the papers were stored in a locked cupboard, and the data from the software was inserted into a flash drive and stored in the same place.

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
