# Peer review of "Symptom Clusters and Mindful Self-Care in People with Cancer in Palliative Care"

_healthcare, 2025, doi:10.3390/healthcare13182317_

Round 1

Reviewer 1 Report

Comments and Suggestions for Authors

The paper addresses significant topic of the improvement of cancer care in palliative care. Despite the relevance of the topic itself the scientific problem is quite ambivalent and presented vaguely in the Introduction. It contains many proclamations and declarative general notions of cancer care, management and importance of mindful self-care for improvement of quality of care? The concept “mindful self-care” is not explained or argued in deep and it’s significance remains quite discussable and not justified at all in the Introduction. How TOUS theory is implicated here? How it is related to the empirical study? What precisely authors mean by aiming “to analyze the relationship between clusters and mindful self-care”? The consistent review of “why it is matter” should be presented clearly and appealing to world-wide practices, not just previous studies in Brasil. The lines 108-116 are not a part of manuscript and probably appeared by mistake.

In Methods it is mentioned that it is cross-sectional study, the criteria of inclusion and exclusion as well as formation of the questionnaire provided. Some major questions occur: was the study anonymous or not, if not, then how confidentiality of research participants was guaranteed, how research participants were involved, if they are severe patients, were they capable to give Informed Consent by themselves and what was the research population number in this care unit and what was response rate. Moreover, how questionnaires were distributed, why data from the pilot study included in the results, and why the adapted scales (like ESA) were used subjectively, i.e. “it was decided not to use the total score of the ten items, but only the individual 148 assessment of each symptom reported by the patient“ (148-149 lines) ? This part also provided too much and not necessarily detailed explanations of instruments (for example, MSCS was previously developed, so it is enough to cite and refer to original authors) and over detailed information about statistical analysis. Paradoxically, results included mainly descriptive data, means and quite complicated interpretation of linear regression. More explicit explanation on the proceeding of this statistical model and the meaning of coefficient numbers would be expected (see 221-227 lines).

The Discussion part some interesting insights and explanations of the presented results, however, again some more text supporting references or citations from other studied are expected (240-242, 252-255 311-314 lines etc.). On other hand, the interpretations of results gained in this study seem to be very general and again declarative, so you rather support or compare your major findings’ data from the statistical analysis with similar studies and you can include in discussion much more publications on Palliative Care.

In conclusions, the more particular highlights of the most important evidences on mindful care impact on symptom management is required. The design of the study, the sample size and other limitations (which were not declared by authors in the manuscript) diminish the importance of this manuscript and requires major revisions and updates. The conclusive statements in the manuscript seem quite general and not really based on the provided findings. Please make conclusions more substantial and related to the results of empirical study Therefore, I would recommend to revise and make conclusions more substantial and related to the results of empirical study. 

Author Response

                                                        RESPONSE LETTER

Dear Editor and Reviewers,

Good morning,

On behalf of our research team, we would like to thank you for all the comments and suggestions provided by the reviewers. We consider all of them highly relevant for the improvement of this research report. Accordingly, we have incorporated all the suggested changes into the text as requested. To facilitate their identification, the revised portions of the text have been highlighted in red.

As we added some of the references suggested by the reviewers and removed those indicated as redundant, it was necessary to revise their order in the manuscript. Consequently, the numbering of the references has been modified and highlighted in green to indicate the updated sequence.

Checklist for Authors

(I) Ensure all references are relevant to the content of the manuscript.

Response: The references have been reviewed and revised as requested by the reviewers.

(II) Highlight any revisions to the manuscript, so editors and reviewers can see any changes made.

Response: All modifications are highlighted in red within the text.

(III) Provide a cover letter to respond to the reviewers’ comments and explain, point by point, the details of the manuscript revisions.

Response: The response letter has been prepared.

(IV) If the reviewer(s) recommended references, critically analyze them to ensure that their inclusion would enhance your manuscript. If you believe these references are unnecessary, you should not include them.

Response: The suggested references have been reviewed and cited in the text.

(V) If you found it impossible to address certain comments in the review reports, include an explanation in your appeal.

Response: The only request not fulfilled concerns the revision by a native English translator. When we simulated this service on the platform, we observed that the final cost would be CHF 330.00, which corresponds to BRL 2,235.47—an expense considered too high by our research team. However, we would like to note that if the study is recommended for publication, we intend to submit it to the MDPI Author Services. Based on the approval, we will be able to request payment through our institution. Would it be possible to proceed in this way?

Once again, we would like to express our gratitude for the opportunity to improve this work.

Sincerely,

The Research Team

Reviewer 2 Report

Comments and Suggestions for Authors

The article presents a study with relevant issues and results concerning a vulnerable population, cancer patients in palliative care. It is well written and organised, so I congratulate the authors on their research and dissemination of knowledge. However, I have some questions and suggestions for improvement.

Abstrat:
•    A paragraph contextualising the study should be developed before the objective;
•    As this is the point of the summary, remove the numerical values of the most frequent symptoms, as well as those of the association between symptoms. They should therefore appear in the presentation of the results.

Introduction:
•    Present bibliographical references on the development of symptoms in cancer patients (lines 52-53);
•    Present the research question or questions of the study;
•    Introducing the concept of health literacy could be interesting in order to later relate it to what is presented in the discussion and conclusions;
•    Lines 108-116 should be removed.

Methodology:
More description and detail regarding the inclusion and exclusion criteria for participants. What treatments were they given? Were these identical for all participants? What follow-up was done with the participants? Were they followed up by a palliative care team from the time of diagnosis?
What references, sources and evidence support the questions asked of participants and supported the assessment for exclusion?

Discussion:
In line 246, regarding the third paragraph, the study refers to diabetic participants, unlike all the others, which also include cancer patients. Therefore, the discussion should be based on the study of cancer patients.
In line 328, it is relevant to develop the concept of health literacy in these patients, as suggested by the study results.
In line 338, aspects related to the limitations of the study should be developed and, specifically, new research should be suggested.

Author Response

(The authors gave the same response as above.)

Reviewer 3 Report

Comments and Suggestions for Authors

This is an interesting study. The phrasing needs work and the English may be improved by a native speaker.
The cluster of symptoms the authors highlight are not attributed to cancer alone, they may be heavily influenced from chemo or radiotherapy, work related fatigue and medications received by the patients. Please provide a detailed use of all of the above in your cohort and correlate them with your findings. 
Also, specific malignancies and overall staging influence these clusters. Please provida all data concerning this and correlate accordingly. 
Also, use the pdf comments provided in the file. 

Comments on the Quality of English Language

The introduction and abstract show very limited English use, while the discussion does not. Please revise. 

Author Response

(The authors gave the same response as above.)

Round 2

Reviewer 1 Report

Comments and Suggestions for Authors

Thank you for your improvements. Few more suggestions to complete the manuscript:

  1. Please revise the whole text again and make sure each sentence is related logically with the next one or add some links in connecting the ideas, for example, the connection of logic in lines 58-59, 72-74, 130-131 etc. By reading the manuscript some gaps in between the sentences could be also detected in the Discussion section.
  2. In amended parts some statements should be supported obviously with the references or cited precisely, especially when authors' names or some specific studies are highlighted, see lines 111, 116, 195, 401, 402, 407-409 etc. 
  3. The new version of conclusions should be more concentrated and providing the major outcomes of your study.
  4. English could be improved in terms of style, terminology and academic language, reformulating the statements in more formal and less daily terms. The usage of articles and caps should be revised too, especially in non amended parts of the manuscript. 
  5. I also recommend to shrink The Results and Conclusions sections in the Abstract. It is not clear why nurse role is highlighted in the conclusive sentence in the Abstract. 
  6. Overall, the whole text should be checked twice and presented tuned up to the academic language by excluding some trivial and excessive statements or off-topic information. 
Comments on the Quality of English Language

English could be improved in terms of style, terminology and academic language, reformulating the statements in more formal and less daily terms. The usage of articles and caps should be revised too, especially in non amended parts of the manuscript. 

Author Response

Dear Editor and Reviewers,

We sincerely appreciate all the comments provided regarding our manuscript. We observed that most of the comments were related to the English writing; therefore, we have submitted the manuscript to the journal’s own language editing service. The other issues, concerning reference citations as well as the revision of the Abstract and Conclusions, have all been addressed as requested.

We are grateful for this opportunity and for your valuable feedback.

Sincerely,

The Authors

Reviewer 3 Report

Comments and Suggestions for Authors

Thank you for your editing. Your results appear clearer now.

Author Response

(The authors gave the same response as above.)
